# Synthetic studies on the tetrasubstituted D-ring of cystobactamids lead to potent terephthalic acid antibiotics
Moritz Stappert[1,2], Daniel Kohnhäuser[1], Tim Seedorf[2], Janetta Coetzee[3], Katharina Rox[1,4], Hazel L. S. Fuchs [1], Katarina Cirnski[3], Christian Leitner[2], Jennifer Herrmann[3], Andreas Kirschning[2,5], Rolf Müller[3,4] & Mark Brönstrup [1,2,4] ✉

Novel scaffolds for broad-spectrum antibiotics are rare and in strong demand because of the increase in antimicrobial resistance. The cystobactamids, discovered from myxobacterial sources, have a unique hexapeptidic scaffold with five arylamides and possess potent, resistance-breaking properties. This study investigates the role of the central D-ring pharmacophore in cystobactamids, a *para*-aminobenzoic acid (PABA) moiety that is additionally substituted by hydroxy and isopropoxy functions. We varied the two oxygenated substituents and replaced both amide connectors with bioisosteres. Synthetic routes were developed that included metal-mediated aromatic functionalization or heterocycle formations, leading to 19 novel analogues. The antibiotic efficacy of all analogues was determined against bacteria from the ESKAPE pathogen panel. While the replacement and the repositioning of hydroxy and isopropoxy substituents was not advantageous, exchanging PABA by terephthalic acid amides led to the highly potent analogue **42** with broad-spectrum activity, insensitivity towards AlbD-mediated degradation and promising pharmacokinetic properties in mice. The study highlights the steep structure-activity relationships in the tetrasubstituted D-ring and a surprisingly favorable reversion of the amide connecting C and D.

Bacterial resistance is omnipresent and expected to be a major future problem as warned by national and global organizations such as the World Health Organisation (WHO)[1–3], or the European Centre of Disease Prevention and Control (ECDC)[4]. Novel scaffolds for broad-spectrum antibiotics that can displace major current classes such as β-lactams or quinolones are rare and their development is slow[5]; however, innovative approaches for antibiotic discovery exist[6]. The cystobactamids and the structurally similar albicidins have an unusual oligoarylamide structure, containing one aliphatic α-amino acid (Fig. 1)[7,8]. Both compound classes proved to exert potent antibiotic activities through the inhibition of bacterial gyrase and topoisomerase IV. The unique binding mode of the oligoarylamides, featuring interactions of its Eastern part with gyrase amino acids, while its Western part binds to DNA, explains why the compounds break resistance towards existing classes of gyrase inhibitors[9]. In first structural optimization studies[7,10–18], the ring D substructure was found to be highly relevant for antimicrobial activity.

For example, an elongation of the isopropoxy to an isobutoxy residue, as well as the removal of the hydroxy function proved to be disadvantageous. However, establishing structure-activity relationships

**Fig. 1 | Structure of reference cystobactamids CN861 (1) and CNCC-861 (2).** The nomenclature of structural elements is shown. The *p*-aminobenzoic acid (PABA) units are alphabetically named. The amide bonds are named after the corresponding connected arenes, for example 'DE-amide bond' is the amide bond between rings D and E.

[1]Department of Chemical Biology, Helmholtz Centre for Infection Research, Braunschweig, Germany. [2]Institut für Organische Chemie und Biomolekulares Wirkstoffzentrum (BMWZ), Leibniz Universität Hannover Schneiderberg 1B, Hanover, Germany. [3]Helmholtz Institute for Pharmaceutical Research Saarland, Universitätscampus E8.1, Saarbrücken, Germany. [4]German Center for Infection Research (DZIF), site Hannover Braunschweig, Braunschweig, Germany. [5]Uppsala Biomedical Center (BMC), University Uppsala, Uppsala, Sweden. ✉e-mail: mark.broenstrup@helmholtz-hzi.de

(SAR) in this part of the molecule has been limited by the higher synthetic challenges associated with the tetrasubstituted arene. In this work, we focused on D-ring analogues starting with the current lead-scaffolds CN861 (**1**) and CNCC-861 (**2**)[19] (Fig. 1). The effects of the re- or displacement of the hydroxy- and isopropoxy functions were assumed to deliver important information about the role of this aromatic ring. In addition, various isosteric groups for the amide bonds were introduced between rings C and D and rings D and E to probe and alter the intramolecular hydrogen bonding network. The in vitro efficacy of the new analogues was determined against various bacteria of the ESKAPE panel, and selected compounds were profiled in specific resistance and pharmacokinetic assays, highlighting their promising antibiotic potential.

## Results

We started with the highly potent compound **2** to introduce variations to ring D. Conspicuously, variations of the alkoxy moiety on ring D frequently occurred for the cystobactamids found in nature, while the hydroxy group on ring D remains unaltered[7,14,20]. For that reason, modifications of the isopropoxy- and hydroxy groups were targeted first. The phenol function can act both as hydrogen bond donor (HBD) and hydrogen bond acceptor (HBA) and may contribute to metabolic stability and solubility[21]. Furthermore it can assume two possible charge states, which influence the interaction with cell membranes and the target itself. The targeted cystobactamid analogues with varied hydroxy and alkoxy residues are depicted in Fig. 2, and the synthesis of the corresponding ring D precursors is shown in Fig. 3.

In hydrochinone **3**, the type of substituents and the electron density of ring D were largely preserved, but the topology of hydroxy as an HBD was rearranged. This was expected to result in a switched conformation of the DE-amide bond and thereby a relocation of the terminal E ring. The protected precursor of ring D **14** was accessed by adapting known protocols (Fig. 3)[22,23]. Starting with commercially available material **12a**, an Elbs-oxidation introduced the hydroxy group *para* to the isopropoxy group. The salicylic acid function was protected as an acetonide, that allowed a selective alkylation of the free hydroxyl group by a Mitsunobu reaction to give **13**. After deprotection and allylation of the 6-hydroxy group, *p*-nitrobenzoic acid **14** was obtained.

Next, the hydroxy group was replaced by halogen atoms as in compounds **4**, **5** and **6**, in order to probe the effect of removing the protic function at this position. The commercially available aldehydes **15a–c** served as substrates for the introduction of the nitro group as fourth substituent. The nitration was sufficiently selective for the desired regioisomers **16a–c**, delivering yields of 38–47%. The site of nitration was verified by 2D-NMR (see the Supporting Information). Pinnick oxidation of the aldehydes, followed by Mitsunobu reactions to introduce the isopropyl groups and subsequent saponifications delivered **17a–c**. This route avoided side reactions of the aldehydes during the Mitsunobu reactions.

To contrast these non-protic phenol isosteres as weak HBA, an amino function as a replacement of the hydroxyl group was targeted, as it is an HBD and a strong +I/ + M substituent. Such anilines are weak bases and known isosteric groups for phenols[24,25]. To obtain cystobactamid **7**, the methyl ester of commercially available **18** was prepared first. An acetyl group was introduced to protect the amino function and to reduce its directing effect during nitration to give **19**. A solution of nitric acid in acetone or DCM turned out to be optimal for nitration, although the yield of up to 27% for this step remained moderate. Compound **19** was isopropylated via a Mitsunobu protocol to give **20**, followed again by mild saponification to **21**. As the removal of the acetyl function was anticipated to require harsh conditions, the formyl-protected intermediate **22** was synthesized as well. Here, the nitration was performed without methyl-protection of the carboxylic acid to give **22** in a yield of 31%. Interestingly, the desired regioisomer was easily obtained by precipitation, while the isolation of **19** required chromatography. After isopropylation of the hydroxy and carboxyl functions, the formyl group and intermediate isopropyl ester were easily cleaved under mild basic conditions to give **24**. Again, the correct site of nitration was verified by 2D-NMR of the products or the final cystobactamids. From this compound the final cystobactamid **7** was obtained. Building block **20** later served as a substrate in the synthesis of a rigidified DE-amide bond in cystobactamids **45a** and **45b**.

We were also interested in the effects of other weak bases as isosteres for the phenol group, such as pyridines. Pyridines as ring D substitutes were recently synthesized[26] and reported to have high activity in albicidins[13]. To test a pyridine on ring D as part of our reference scaffold **2**, compound **8** was targeted next. For this purpose **27** was synthesized according to the literature[26]. Here the Stille coupling of **26** with a vinyl reagent was

**Fig. 2 | Ring D analogues (part 1).** modifications of the oxygen-substituents synthesized in this work.

**Fig. 3 | Synthesis of building blocks carrying ring D substitutions.** Reagents and conditions: (a) $K_2S_2O_8$, NaOH, 4 d[5]; (b) acetone, TFA/TFAA[6]; (c) iPrOH, DIAD, PPh$_3$; (d) NaOMe[5]; (e) AllBr, $K_2CO_3$; (f) LiOH, RT-60°C; (g) HNO$_3$ (fum.), ACN, 0°C; (h) NaClO$_2$, buffer solution, 2-Me-2-butene; (i) MeOH, $H_2SO_4$, reflux, 1-2 d; (j) Ac$_2$O, Py; (k) HNO$_3$ (fum.), acetone, 0°C; (l) HCOOH, Ac$_2$O; (m) HNO$_3$, DCM, -40°C; (n) iPrOH, NaH, (o) vinyl-BF$_3$K, Pd(PPh$_3$)$_4$, $K_2CO_3$, $H_2O$/dioxane, 100°C; (p) KMnO$_4$/ acetone; (q) SOCl$_2$, MeOH, 80°C; (r) iPrBr, NaH, DMF; (s) NBS, AIBN, CCl$_4$, 100°C; (t) NMO; (u) DAST, DCM; (v) LiAlH$_4$, THF; (w) PCC; (x) tBuOH, DCC, DMAP; (y) SnCl2; (z) MOMCl, NaH, DMF; (a2) MeCOCl, Py; (b2) tBuLi, THF, -78°C to -20°C, then CuCN*LiCl, methallyl bromide, -78°C to -20°C; (c2) Pd/C, H2, MeOH; (d2) TMSCl, MeOH; (e2) KOH (4 M), reflux; (f2) Boc$_2$O, EtOH; (g2) (CH$_2$O)$_n$, MgCl$_2$, TEA, ACN/DMPU, reflux.

successfully replaced by applying less toxic Suzuki coupling conditions. Oxidation of the vinyl pyridine by KMnO$_4$ gave carboxylic acid **27**.

Analogue **9** bears a difluoromethyl group that is isosteric for the hydroxy function and is an HBD as well[27]. To obtain its ring D fragment **32** starting from 2-methyl-3-hydroxybenzoic acid **28**, the methyl ester was prepared first, followed by isopropylation of the hydroxy function. A radical monobromination of the benzylic methyl group gave **29**. To introduce the difluoromethyl function, the benzyl bromide was first oxidized to the aldehyde using N-methylmorpholine N-oxide. Subsequent conversion with

the fluorinating agent DAST gave **30**. The direct nitration with this material was unfortunately unsuccessful. However, after reducing the ester to aldehyde **31**, the nitration worked and gave the desired regioisomer in 23% yield. After oxidation of the aldehyde under Pinnick conditions, **32** was obtained.

In addition to the presented isosteres for the hydroxy group, substitutions of the isopropoxy function on ring D were investigated. In our previous work, the replacement of isopropoxy by isobutoxy led to a decrease in activity[10]. However, methoxy as a substituent displayed activity in some naturally occurring cystobactamids and in albicidins[8,14]. We were interested

**Fig. 4 | Ring D analogues (part 2).** Replacements for amides flanking ring D synthesized in this work.

how this substructure would perform when combined with our reference scaffold **2**. Targeting cystobactamid **10**, compound **34** was obtained from **33**, which in turn was prepared according to literature[15].

As another modification the formal substitution of the ethereal oxygen of **2** by methylene as in **11** was targeted. This replacement would shed light on the role of the oxygen atom as an HBA and thus its interaction with the adjacent amide and hydroxy functions; it would also reduce electron density in the D-ring. The synthesis of the corresponding building block **39** was attempted via a directed *ortho*-metalation of **36**. The latter was synthesized from **35** by adapting known protocols that however used carboxy functions such as dimethylformamide and chloroformate as electrophiles[28,29]. In order to introduce an alkyl chain instead, we adjusted the reaction conditions and used methallyl bromide as electrophile and a cyano cuprate as co-reactant. Such cuprates are known to facilitate the conversion of alkyl and alkenyl halides in lithiation reactions[30–33]. Indeed, the turnover to the desired product was increased by 50% compared to cuprate-free conditions. Isobutyl bromide, was replaced by methallyl bromide in order to avoid elimination reactions obtained with the former.

After hydrogenation of the isobutenyl group, the MOM protecting group was removed to give **37**. Unfortunately, the Moc protecting group turned out to be resistant towards removal using strong nucleophiles as propane thiolate and TMS-iodide at a later synthetic stage. Therefore, a de-/reprotection with Boc became necessary. The last ring D-substituent was then introduced by a Skattebol-formylation *ortho*-selective to hydroxy. By systematic optimization of the reaction conditions, aldehyde **38** was obtained in an acceptable yield of 39% over three steps. After allyl protection and a Pinnick oxidation of the aldehyde, carboxylic acid **39** was obtained. In

the latter, the oxidation protocol was improved by substituting the usual scavenger 2-Me-2-butene with $H_2O_2$, thus circumventing hardly removable organic residues in the product.

## Synthesis of amide isosteres attached to ring D

The second set of analogues featured eight modifications of the amide groups attached to ring D (Fig. 4).

A benzylamine function between rings C and D as in **40** was targeted, because it adds rotational flexibility to the amide backbone and removes oxygen as an HBA. The synthesis started from 4-nitrobenzaldehyde **49** and the previously prepared DE fragment **50** (Fig. 5). Sodium triacetoxyborohydride, a mild reducing reagent that was prepared in situ, was used for the reductive amination. The coupling proceeded slowly, however, the conversion was finally increased by repeatedly working up the reaction solution, isolating the reactant/product mixture and adding reagents again. The obtained product mixture was subjected to Boc-protection, which led to a product mixture. CDE-fragment **51** was finally obtained in a yield of 9.2% over three steps.

The benzylamine moiety was also introduced between rings D and E to give cystobactamid **41**. The applied conditions were similar to those applied for **51**, but the amination reaction was faster and gave significantly higher yield.

Next, the synthesis of cystobactamid **42** was targeted, a terephthalic acid analogue formally obtained by a reversion of the CD-amide linker. This change was expected to increase the H-bond strength between the amide and the adjacent isopropoxy function and can thus stabilize alternative conformations (see SAR discussion for details). The synthesis was established from commercially available 2,3-dihydroxyterephthalic acid **56**,

**Fig. 5 | Synthesis of DE- and CDE-fragments carrying amide replacements.**
Reagents and conditions: (a) NaBH$_4$, AcOH (3 Eq), DCM; (a2) **55**, NaBH$_4$, AcOH (3 Eq), DCM; (b) Boc$_2$O, DMAP; (c) Zn/AcOH; (d) AcCl or TMSCl, MeOH; (e) iPrOH (1 Eq), DIAD, PPh$_3$; (f) K$_2$CO$_3$, nicotinic acid; (g) *tert*-butyl 4-aminobenzoate, P(OPh)$_3$, toluene, reflux; (h) AllBr, K$_2$CO$_3$; (i) LiOH; (j) 4-nitroaniline, T3P, Py; (k) Br$_2$, *t*BuNH$_2$, toluene/DCM; (l) Ac$_2$O, Py; (m) HNO$_3$, DCM, -40°C; (n)

ethynyltrimethylsilane, PdCl$_2$(PPh$_3$)$_2$, CuI, DiPEA; (o) K$_2$CO$_3$, MeOH; (p) 4-azidobenzoic acid, CuSO$_4$, Na-ascorbate, THPTA; (q) 4-nitrobenzoyl chloride, Py, DCM; (r) **72**, K$_2$CO$_3$, Pd(PPh$_3$)$_4$, dioxane/H$_2$O; (s) MOM-Br, DMAP, DiPEA; (t) 4-amino *t*Bu benzoate, (COCl)$_2$ or POCl$_3$, DMF (cat.), DCM, Py; (u) HCOOH, Ac$_2$O; (v) EDC, DMAP, *t*BuOH; (w) **82**, POCl$_3$, DiPEA, DCM; (x) +**52**, NH$_4$OAc, air, H$_2$O; (y) Na$_2$S$_2$O$_4$, EtOH/H$_2$O; (z) **81**, air; (a2) allyl chloroformate, DiPEA, 0°C.

which was converted to the dimethyl ester under acidic conditions in methanol. After monoisopropylation using Mitsunobu conditions, the ester groups were differentiated by a regiospecific demethylation with nicotinic acid and potassium carbonate in DMF to give **57**. The free carboxylic acid was amidated with *tert*-butyl 4-aminobenzoate using triphenyl phosphite at higher temperature. After allyl protection and saponification of the methyl ester, **58** was obtained as a free carboxylic acid. Amide coupling with 4-nitroaniline, facilitated by T3P, and subsequent reduction of the nitro group by Zn/AcOH afforded **59**.

We now concentrated on the amide bond between ring D and E, which was ascribed an important role in conformer stabilization[26]. Since the adjacent hydroxy group is both an HBD and an HBA, at least two conformations were found to be energetically possible. In order to identify the conformation relevant for bioactivity, we introduced amide isosteres and bicyclic systems with fixed conformations. Cystobactamid **43** replaced the amide with a triazole group. The amide was completely omitted in **44** to give a non-planar biaryl function of rings D and E. We expected that this change would reduce the π-π stacking and therefore increase solubility. Both analogues were accessible by cross coupling reactions with arylbromide **62**. This compound was obtained after regioselective bromination of commercially available **60**. Acetylation and a regioselective nitration gave **62** in a yield of 78% over two steps. Again, the correct regioisomer was identified via 2D-NMR. Compound **62** was first converted to alkyne **63** under Sonogashira conditions, followed by basic hydrolysis and copper-catalyzed azide-alkyne cycloaddition (CuAAC) with 4-azidobenzoic acid as ring E building block. After global protection with allyl bromide and reduction of the nitro group, **64** was obtained. The full CDE-fragment **65** was obtained after amide coupling with 4-nitrobenzoic acid, followed by reduction under the known conditions.

The biphenyl system of **44** was introduced from **62** by a Suzuki-coupling with pinacol borane **72**. Hydrolysis of the acetate protecting group followed, giving **66** in 39% yield over two steps. After protection of the hydroxy group with MOM and reduction of the nitro group, **67** was obtained. **68** was then prepared via two steps analogously as seen before.

As mentioned above, *N*-acetylated compound **69** was originally prepared as an intermediate towards an amino substituted ring D. However, the amide coupling between **69** and 4-amino *t*Bu-benzoate, facilitated by (COCl)$_2$ and pyridine, gave **70** through a heterocyclization. After nitro reduction, coupling with 4-nitrobenzoyl chloride and renewed nitro reduction, CDE fragment **71** was obtained. The methyl oxoquinazoline structure was identified via LCMS and NMR, and regarded as a welcome addition for the SAR investigation with its rigidified DE-system, giving cystobactamid **45a** as the final compound.

To determine the effect of the methyl group in the oxoquinazoline ring of **45a**, desmethyl analogue **45b** was prepared. The corresponding CDE fragment **75** was prepared starting from the formylation of already available **73** and coupling of the product with 4-amino *t*Bu-benzoate under similar heterocyclization conditions as described before. Interestingly, the cyclization did not occur with DiPEA, but required pyridine as activating base. Nitro reduction gave compound **74**, which was further assembled to the final cystobactamid via **75**.

The rigidifications of **45a** and **45b** were placed between ring D and the DE-amide bond. We now expanded this to rigidified structures between the DE-amide bond and ring E: The quinazoline system of cystobactamid **47** would introduce two HBAs that allow interaction with the hydroxy substituent at ring D as HBD. This in turn would lead to the stabilization of two conformers with coplanar rings D and E.

To obtain DE-fragment **78**, synthesis started from commercially available **76**. After nitro reduction to the corresponding amine, the reaction with aldehyde **52** followed. This ammonium acetate-facilitated reaction produced the quinazoline ring after oxidation under air in aqueous medium. The desired product **78** was obtained from **77** in 14% yield, which was due to unavoidable side reactions of the reducing reagent with the quinazoline.

We also considered introducing a rigidification between the DE-amide bond and ring E that, in contrast to **47**, would offer both HBA and HBD functionalities. Therefore, benzimidazole **48** was chosen as the next target,

whose synthesis started with a condensation of aldehyde **52** with methyl 3,4-diaminobenzoate **81**. The oxidative formation of imidazole **79** proceeded under air. Alloc-protection of the imidazole and nitro reduction followed, producing two regioisomers **80** in good yield. In contrast to **78**, the heterocycle was inert during the reduction, reflecting the higher electron density of benzimidazoles compared to quinazolines.

With the last derivative **46** we wanted to find out the effects of removing the HBD-properties of the DE amide by methylating it. The *N*-Me PABA **82** was first *t*Bu-protected via an EDC/DMAP mediated esterification. Amide coupling of the secondary amine with **85** and subsequent nitro reduction gave **83**. The CDE-fragment **84** was obtained analogously as seen before.

## Assembly strategy

The CDE-fragments shown in Fig. 5 had to be assembled to the full cystobactamids. Several total syntheses of cystobactamids were reported that differ in the order of assembling the ring elements[10,14,16,17]. In our recent work, the last disconnection was made between ring B and the central amino acid (Fig. 6, method 1)[19].

However, for synthetically valuable ring D analogues, a further reduction of steps to the final cystobactamids was considered important. Therefore, we have set the last disconnection between the central amino acid (CAA) and ring C (method 2). This allows direct amide coupling of Western and Eastern fragments. By switching to this updated synthetic strategy, two reaction steps involving fragments with elaborated ring D building blocks were saved.

The synthesis of the corresponding AB-fragment and the following assembling of **5** as an example cystobactamid is depicted in Fig. 6. AB-fragment **89** was synthesized as reported[34]. Coupling with CAA **94** required the protection of the carboxylic acid by the formation of methyl ester **95** under mild conditions with TMSCl in methanol. After HATU-facilitated amide coupling, the methyl ester was cleaved under basic conditions, and fragment **92** was obtained. Assembly of this fragment with the CDE-fragments was established by T3P-mediated amide coupling. A global deprotection concluded the synthesis of the full cystobactamids. The overall yields are mentioned in Tables 1 and 2.

## Evaluation of antibacterial properties

All cystobactamids were evaluated in MIC-assays on a panel of representative ESKAPE strains with focus on *A. baumannii* (Tables 1 and 2). Compared to our previous work[10,17,18], we specifically included more challenging test strains with known resistances against gyrase inhibitors.

In summary, substitutions of the hydroxy group at ring D were consistently associated with a severe loss of activity. The shift of the hydroxy group from the 2- to the 6-position of ring D led to an almost complete loss of activity in **3**, with residual activity against *E. coli* and *A. baumannii* strains. The substitution of the hydroxy group by fluorine in **4** led to a similar activity pattern, while substitutions with chlorine and bromine as in **5** and **6**, respectively, abolished the activity in all tested strains. The amino-substituted cystobactamid **7** showed only residual activity against the *E. coli* reference strain and a clinical isolate of *S. pneumoniae*. The pyridine derivative **8** exhibited a comparably low activity, which is in some contrast to the reported significant activity towards selected bacterial strains of albicidins with comparable nitrogen insertions[13]. Also a difluoromethyl substitution for hydroxy as in **9** led to complete elimination of activity. One can conclude that (1) the substitution of the hydroxy group with groups that are exclusively HBAs is not tolerated, (2) an HBD as a hydroxy-substitute is not sufficient on its own and (3) a phenolic hydroxy group is probably not interchangeable here due to its unique H-bonding pattern and acidic properties.

The substitution of the isopropoxy group of **2** with a methoxy function as in **10** mostly decreased the activity, with potency being retained against some *A. baumannii, E. coli* and *S. aureus* and *S. pneumoniae* strains. However, the isopropoxy replacement had a smaller effect on activity than the substitutions of the hydroxyl group. The substitution of oxygen as an HBA in the isopropoxy group chain with methylene as in **11** strongly

**Fig. 6 | Late-stage retrosynthetic disconnections for cystobactamid 2.** Previously reported procedure (method 1) and updated method 2 with exemplified synthesis of cystobactamid **5** are shown. Reagents and conditions: (a) DMA as solvent; (b) **95**, HATU, DiPEA; (c) LiOH, THF/H$_2$O; (d) TMSCl, MeOH; (e) **93**, T3P, Py; (f) TFA, DCM.

decreased activity against several species. Residual activity was observed for some *A. baumannii* strains, the *E. coli* reference strain and *E. faecium*.

To investigate the SAR on the amide isosteres flanking ring D (Fig. 4), the MICs were determined on the same pathogen panel as seen before (Table 2). The insertion of methyleneamine as linker between ring C and D as in **40** reduced activity compared to **2**, although some *A. baumannii, E. coli* and *E. faecium* strains remained sensitive, in a pattern that was similar to the pattern of the isobutyl derivative **11**.

In contrast to this, a reversion of the amide between rings C and D was beneficial: The terephthalic acid derivative **42** showed exceptional activity that even exceeded that of the reference **2**. It also had higher activity than ciprofloxacin against *K. pneumoniae* and *A. baumannii* in particular.

Compounds **41, 43**, and **44**, which contain isosteres for the DE-amide, were all inactive. Similar to the hydroxy function, any isosteric substitutions of the DE-amide do not appear to be tolerated. This is a pronounced contrast to the CD-amide, where the corresponding methyleneamine **40** and triazol substitutions were tolerated[10].

Residual activities, limited to some non-fluoroquinolone resistant *A. baumannii* strains, the *E. coli* reference strain, as well as the gram-positive species *E. faecium* and the *S. aureus* reference strain, were seen for the two oxaquinazoline derivatives **45a** and **45b**, without a recognizable influence of the methyl group. The loss of activity for **47** and **48** was explicit. The aimed-for stabilizing effect of a nitrogen heterocycle as isostere for the DE-amide could not be shown.

Also the methylated analogue **46** was inactive, with the exception of the sensitive *E. coli* type strain ATCC 25922. These findings underline the high relevance of an amide with HBD capability connecting rings D and E.

## Conclusions for SAR

For all discussions of the SAR, it must be taken into account that gyrase sequences may differ substantially between bacterial species and thus the molecular target interactions may not be the same across species. This makes it difficult to predict broad spectrum activity only based on ligand data. Irrespective of this, the possible conformations that are relevant for

**Table 1 | Overall yields and antibacterial activity of cystobactamid D-ring analogues (part 1): substitutions for hydroxy and isopropoxy groups, given as minimal inhibitory concentrations (MICs) in µg mL⁻¹**

| | CN-CC-861 2 | CN861 1 | Cmpd. 3 | Cmpd. 4 | Cmpd. 5 | Cmpd. 6 | Cmpd. 7 | Cmpd. 8 | Cmpd. 9 | Cmpd. 10 | Cmpd. 11 | CIP |
|---|---|---|---|---|---|---|---|---|---|---|---|---|
| Overall yield | - | - | 1.6% 15 steps | 6.6% 9 steps | 5.9% 9 steps | 3.6% 9 steps | 2.9% 10 steps | 27% 8 steps | 17 steps | 15% 8 steps | 2.8% 16 steps | - |
| A. baum. DSM-30008 [a] | 0.02–0.03 | 1 | 1 | 0.06 | >8 | >8 | n.d. | n.d. | >8 | n.d. | 1 | 0.05–0.25 |
| A. baum. ATCC BAA-1710 | 0.06–0.25 | 8 | >64 | >8 | >8 | >8 | >64 | >64 | >8 | 0.25 | 0.25 | 32 |
| A. baum. CIP-105742 | 0.02 | 0.25 | 2 | 0.25–0.5 | >8 | >8 | n.d. | n.d. | >8 | n.d. | <0.03 | <0.03–0.25 |
| A. baum. CIP-107292 | 0.5–1 | >64 | >64 | >8 | >8 | >8 | >64 | 8 | >8 | 4 | >64 | 64 |
| A. baum. R835 | 1 | 64 | >64 | >8 | >8 | >8 | >64 | >64 | >8 | 4 | >64 | 32 |
| E. ae. CIP 106754 | 0.125–4 | 4 | >64 | >64 | >64 | >64 | >64 | >64 | >64 | 4 | >64 | >6.4 |
| E. cloacae ATCC BAA-2468 | 0.06–0.5 | 32 | >64 | >64 | >64 | >64 | >64 | >64 | >64 | 16 | >64 | >6.4 |
| E. coli ATCC-25922 [a] | 0.03–0.06 | 0.25 | 1 | 4 | >64 | >64 | 0.005 | ≤0.03 | >8 | <0.03 | <0.03 | 0.01–0.03 |
| E. coli LM705 | <0.03–0.2 | 0.64 | >64 | >64 | >64 | >64 | >64 | >64 | >64 | <0.03 | >64 | >6.4 |
| K. pneum. CIP-104298 [a] | 8 | 16 | >64 | >64 | >64 | >64 | n.d | n.d. | >64 | n.d. | >64 | 0.025–0.05 |
| K. pneum. KP10581 | 0.06–0.25 | 2 | >64 | >64 | >64 | >64 | >64 | >64 | >64 | 1 | 8 | >6.4 |
| K. pneum. R-1525 | 16– > 64 | 16 | >64 | >64 | >64 | >64 | >64 | >64 | >64 | 32 | >64 | >64 |
| P. ae. PA14 [a] | 0.5–16 | 1 | >64 | >64 | >64 | >64 | n.d. | n.d. | >64 | n.d. | >64 | 0.025–0.2 |
| P. ae. PA14ΔmexAB | 0.03–0.5 | n.d. | >64 | >64 | >64 | >64 | n.d. | n.d. | >64 | n.d. | >64 | 0.006–0.1 |
| E. faec. DSM-17050 (VRE) | <0.03 | n.d. | >64 | 4 | >64 | >64 | >64 | 0.25 | >64 | n.d. | <0.03 | 0.05–4 |
| S. aur. ATCC-29213 [a] | 0.02 | 0.125 | >64 | >64 | >64 | >64 | >64 | >64 | >64 | 0.5 | 16–32 | 0.2–1 |
| S. pneum. DSM-11865 (PRSP) | 0.125 | n.d. | >64 | >64 | >64 | >64 | 0.2 | n.d. | >64 | <0.03 | >64 | >6.4 |

Reference compounds: CN-CC-861 2, CN 861 1, CIP (= ciprofloxacin).
A. baum. = Acinetobacter baumannii; E. ae. = Enterobacter aerogenes; E. cloacae = Enterobacter cloacae; E. coli = Escherichia coli; K. pneum. = Klebsiella pneumoniae; P. ae. = Pseudomonas aeruginosa; E. faec. = Enterococcus faecium; S. aur. = Staphylococcus aureus; S. pneum. = Streptococcus pneumoniae. a) reference strain, CIP = Ciprofloxacin, n.d. = not determined.

**Table 2 | Overall yields and antibacterial activity of cystobactamid D-ring analogues (part 2): substitutions for the adjacent amide groups, given as minimal inhibitory concentrations (MICs) in µg mL⁻¹**

| Overall yield | CN-CC-861 2 - | CN861 1 - | Cmpd. 40 6.4% 6 steps | Cmpd. 41 20% 10 steps | Cmpd. 42 7.7% 11 steps | Cmpd. 43 5.0% 12 steps | Cmpd. 44 2.1% 13 steps | Cmpd. 45a 11 steps | Cmpd. 45b 3.9% 10 steps | Cmpd. 47 6 steps | Cmpd. 48 12 steps | Cmpd. 46 10 steps | CIP - |
|---|---|---|---|---|---|---|---|---|---|---|---|---|---|
| A. baum. DSM 30008 [a] | 0.02–0.03 | 1 | 0.5 | >8 | 0.125 | >8 | >64 | 8 | 1–2 | >64 | 8 | >8 | 0.05–0.25 |
| A. baum. ATCC BAA-1710 | 0.06–0.25 | 8 | 0.5 | 1–2 | <0.03–0.05 | >8 | >64 | 0.5–1 | >8 | n.d. | >8 | >8 | 32 |
| A. baum. CIP-105742 | 0.02 | 0.25 | 0.25 | >8 | <0.03 | >8 | >64 | 4 | 2 | >64 | >8 | >8 | < 0.03–0.25 |
| A. baum. CIP-107292 | 0.5–1 | >64 | >8 | >8 | 0.05 | >8 | >64 | > 8 | >8 | >64 | >8 | >8 | 64 |
| A. baum. R835 | 1 | 64 | >8 | >8 | 0.5 | >8 | >64 | > 8 | >8 | >64 | >8 | >8 | 32 |
| E. ae. CIP 106754 | 0.125–4 | 4 | >64 | >64 | 1 | >64 | >64 | > 64 | >64 | n.d. | >64 | >64 | >6.4 |
| E. cloacae ATCC BAA-2468 | 0.06–0.5 | 32 | >64 | >64 | 0.25 | >64 | >64 | > 64 | >64 | n.d. | >64 | >64 | >6.4 |
| E. coli ATCC 25922 [a] | 0.03–0.06 | 0.25 | 0.5–1 | >64 | <0.03 | >64 | >64 | 2–4 | 1 | n.d. | 64 | <0.03 | 0.01–0.03 |
| E. coli LM705 | <0.03–0.2 | 0.64 | >64 | >64 | 0.06 | >64 | >64 | > 64 | >64 | >64 | >64 | >64 | >6.4 |
| K. pneum. CIP-104298 [a] | 8 | 16 | >64 | >64 | <0.03 | >64 | >64 | > 64 | >64 | >64 | >64 | >64 | 0.025–0.05 |
| K. pneum. KP10581 | 0.06–0.25 | 2 | 1 | >64 | <0.03 | >64 | >64 | 4 | 16 | >64 | 16 | >64 | >6.4 |
| K. pneum. R-1525 | 16–>64 | 16 | >64 | >64 | <0.03 | >64 | >64 | >64 | >64 | >64 | >64 | >64 | >6.4 |
| P. ae. PA14 [a] | 0.5–16 | 16 | >64 | >64 | 4 | >64 | >64 | >64 | >64 | n.d. | >64 | >64 | 0.025–0.2 |
| P. ae. PA14ΔmexAB | 0.03–0.5 | 1 | 2 | >64 | <0.03 | >64 | >64 | 16–32 | >64 | >64 | >64 | >64 | 0.006–0.1 |
| E. faec. DSM 17050 (VRE) | <0.03 | n.d. | 0.25 | 16–32 | <0.03 | >64 | <0.03 | 2–4 | 1 | n.d. | 2 | >64 | 0.05–4 |
| S. aur. ATCC 29213 [a] | 0.02 | 0.125 | 0.5–1 | 16–32 | 0.06 | >64 | >64 | 4 | 8 | >64 | 4 | >64 | 0.2–1 |
| S. pneum. DSM 11865 (PRSP) | 0.125 | n.d. | >64 | >64 | <0.03 | >64 | >64 | >64 | >64 | n.d. | >64 | >64 | >6.4 |

Reference compounds: CN-CC-861 **2**, CN 861 **1**, CIP (= ciprofloxacin).
A. baum. = Acinetobacter baumannii; E. ae. = Enterobacter aerogenes; E. cloacae = Enterobacter cloacae; E. coli = Escherichia coli; K. pneum = Klebsiella pneumoniae; P. ae. = Pseudomonas aeruginosa; E. faec. = Enterococcus faecium; S. aur. = Staphylococcus aureus; S. pneum. = Streptococcus pneumoniae. a) reference strain, CIP = Ciprofloxacin, n.d. = not determined.

**Fig. 7 | Nomenclature for conformations of the CDE-fragment used in this work: The amide 'backbone' is highlighted.** The conformers are named after the orientation of the carbon-nitrogen bond (red) of CE- and DE-amide linkers relative to the isopropoxy group. Here, *a* and *d* indicate approximate and distant orientation.

activity can be discussed from the obtained MIC data. These conformations are stabilized by intramolecular H-bonds (IMHBs) as they occur frequently in ring D.

In the following discussion, a coplanar orientation of the aryl units in the cystobactamid Eastern fragment is assumed. The corresponding conformers are stabilized by IMHBs as shown in Fig. 7 and Fig. 8. With respect to the Eastern fragment and especially ring D, we propose a differentiation in antiperiplanar (*a-d* and *d-a*) and synperiplanar (*a-a* and *d-d*) conformations.

For the reference cystobactamid **2** the *d-d* and *d-a* conformations are most relevant with regard to H-bonding. The assumed 3D-structure of the *d-a* conformation is depicted, featuring tetrahedral coordination of the isopropoxy oxygen with the CD-amide and the hydroxy proton. This conformation has the highest number of H-bonds of all possible conformers with coplanar benzene rings. The result is a rotation of the alkyl chain out of the aromatic plane. The same conformeric description is also valid for compound **10** with a methoxy substituent on ring D. Cryo-EM structures of the albicidin Albi-1 in complex with *E. coli* gyrase reproduce this antiperiplanar *d-a* conformation, but show partial non-coplanar orientation of the aryl units in the Eastern fragment[9]. In the hydroquinone derivative **3**, the *d-d* conformation now corresponds to the *d-a* conformation in **2** and vice versa. However, the H-bond between the hydroxy and isopropoxy functions in **2** is missing in **3** due to the *para*-substitution pattern. This destabilization might contribute to the inactivity of **3**. It also implies a reduced relevance of the *d-d* conformation.

By replacing the hydroxy group with halogens as fluorine in compounds **4**, **5** and **6**, the HBD and with it a possible H-bond with the ethereal oxygen is removed. An interaction of fluorine with the adjacent amide proton of the DE-linker is still possible (resulting in the *d-a* conformation) but is assumed to be weak compared to oxygen[35]. For chlorine and bromine, this conformation is less stable.

The amino substitution of the hydroxy function in compound **7** is assumed to result in the *d-d* conformation. A population of the *d-a* conformation is not expected due to sterical hindrance, even if the geometry of the amino function in aniline is slightly pyramidalized and not planar[36].

The most relevant conformation of the pyridine derivative **8** bear an IMHB that forms a five-membered cycle. Compared to the H-bond in the corresponding six-membered cycle of the previously seen derivatives, the H-bond is probably weaker here. A five-membered ring can also be formed with a protonated pyridine that donates an H-bond to the amide oxygen, leading to the *d-d* conformation.

In compound **9** an HBD is available[27], but due to steric repulsions only the *d-d* conformation seems reasonable. Together with the results found for **7**, this conformation appears to be disfavored.

Since the elimination of the hydroxy function affects many different bacteria, this molecular site seems to be part of a common mechanism of action shared by many species. In cryo-EM structural data of a DNA gyrase complex with the albicidins and cystobactamids (resolution: 2.6 Å), H-bond interactions of the hydroxy group with the gyrase binding site are not visible, as the phenolic hydroxy group and the nearby α-helices of the gyrase subunits are too far apart to enable an interaction[9]. This would imply that the hydroxy group is not primarily important for the interaction with the target, but for intramolecular H-bonding. It may also play a role in permeation into the bacterial cell.

In compound **11** the hydroxy group is present, but there is no ethereal HBA. Therefore, the number of possible H-bonds is reduced. In principle, all coplanar conformers are possible here. Sterical hindrance between the isobutyl residue and the oxygen of the CD-linker occur with the *a-a* and *a-d*-, but not with the *d-d* and *d-a* conformers. It can be concluded that replacing the isopropoxy oxygen as HBA with a methylene group considerably enlarges the conformational space, and that the ethereal hydrogen bonds contribute to bioactivity.

A similar argumentation is possible for compound **40** which has a rotatable bond in the connection between ring C and D. Even if the H-bond pattern corresponds to that of **2**, the DE-fragment can now adopt many more conformations. The benzylamine is expected to be a weaker HBD than amide, according to reported structural group constants[37].

In previously discussed compounds, the IMHB between the proton of the CD-linker and the isopropoxy oxygen led to a five-membered ring geometry. In **42**, a six-membered ring geometry can be established, which would result in an increased angle between donor-hydrogen and acceptor of ~120°. With known H-bonds, such angles are usually greater than 150°, although ethers as HBA also allow smaller angles[38]. For this reason the H-bond in **42** might be stronger than in **2**.

The most relevant conformers are therefore the *a-d-* or the *a-a* conformation, with three H-bonds formed in the latter. Compared to the *d-a* conformation of reference compound **2**, this conformation results in a different orientation of ring C relative to ring D. The superior activity of **42** compared to all other compounds suggests that stabilization of these conformations is highly beneficial for activity. Interestingly, the *a-a* conformation was not observed yet in the albicidin-gyrase-DNA complex[9] or in a conformational analysis of the cystobactamid CDE-fragment[26]. Another aspect is that the aromatic system on ring D is more electron-deficient in **42** than in the natural cystobactamids and albicidins. This in turn may substantially lower the pK$_A$ of the phenol group and thus influence bonding interactions.

Due to the missing oxygen in the DE-linker of compound **41**, stabilization of a *d-d* conformation is not expected here. Moreover, the HBD strength is lower, and the rotational flexibility is higher compared to **2**. These factors at the pharmacophoric hotspot of the molecule probably contribute to the observed loss of activity. The *d-d* conformation is also the most stabilized conformation for compound **43**. A second topological feature of **43** concerns the central axis of ring E, that is not parallel to the central axis of ring D, as in natural analogues. This will result in a different positioning of the terminal carboxylic acid function. The lack of bioactivity suggests that any of the two factors are unfavorable.

The rigidification between rings D and E by introducing oxoquinazolines in compounds **45a** and **45b** replaced the hydroxy group as HBD by a nitrogen substituent, which acts only as an HBA here. This now fixed *d-a* conformation, implied as being important by comparing **2** and **3**, is obviously not sufficient to confer activity.

The *N*-methylated amide group between rings D and E in compound **46** cannot act as HBD, so that only the *d-d* conformation is reasonably stabilized. Together with the results from **3**, **7** and **9**, the data suggest that this conformation is not relevant for activity.

The direct connection of ring D to E with a biphenyl system in compound **44** was not tolerated and led to a complete loss of activity. This might result from the loss of coplanarity of both aromatic systems[39], but also the shortened distance from ring E to the carboxylic acid of ring E might contribute. A hint is given by **47**, where coplanarity of the biaryl system is favored by a hydrogen bond between the phenol and the nitrogen of the quinazoline. The absence of bioactivity in **47** and in **48** suggest that coplanarity alone is not sufficient for activity.

**Fig. 8 | Possible conformations of the CDE-fragment in the tested compounds.** The conformers shown were selected because they have the largest possible number of H-bonds (dashed) that stabilizes the corresponding conformer, regardless of the conditions in the binding pocket. Disfavored conformers are also shown if they are important. The amide 'backbone' is highlighted in blue. 3D-structures were made with *Avogadro*.

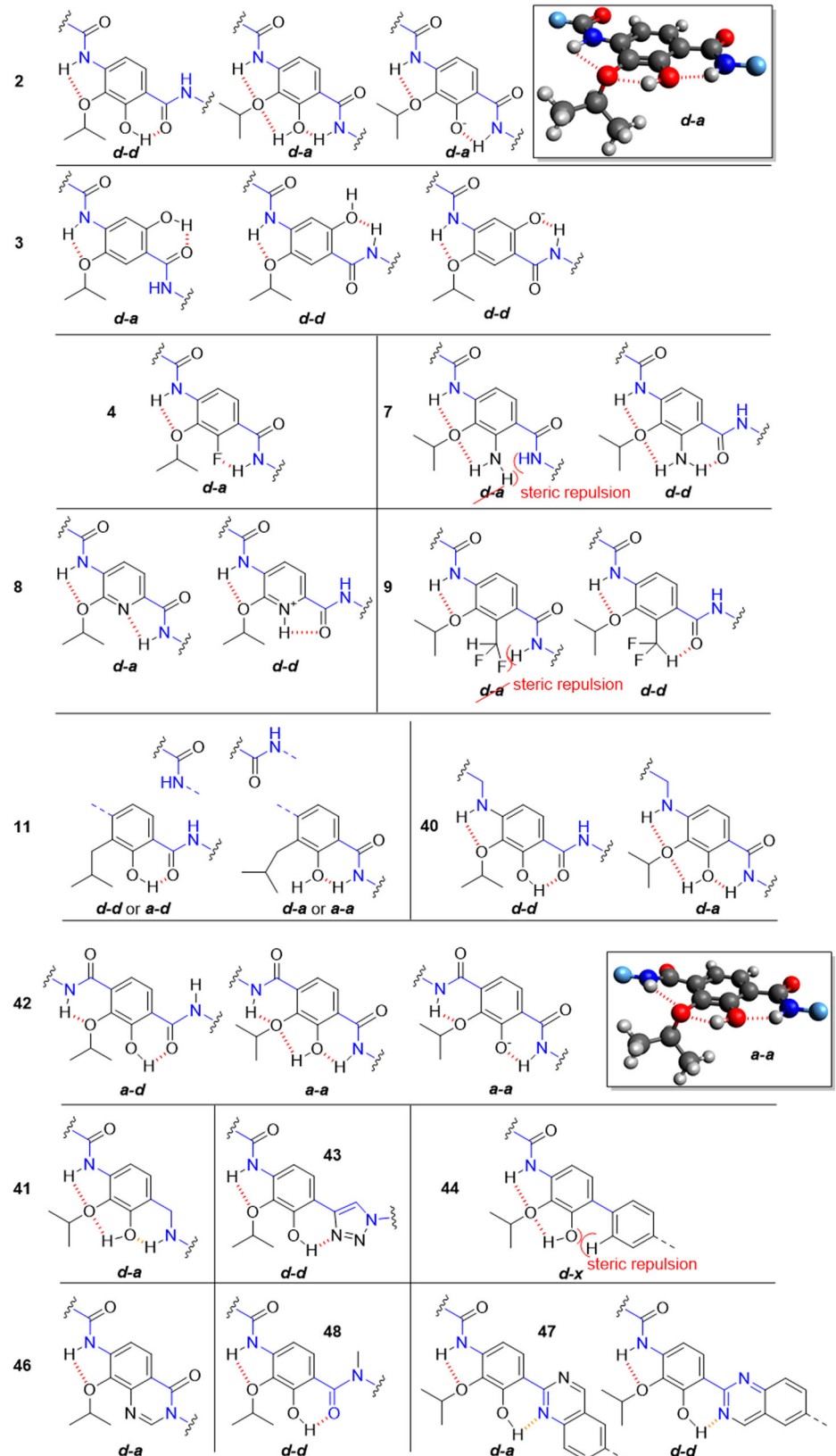

In earlier studies computational structural analysis of the eastern fragments was conducted for cystobactamids and albicidins in solvent without target interaction. Some of these data suggest the *d-d* conformation[13,26], some the *d-a* conformation[40] as the most stabilized. The latter was also observed in the earlier mentioned cryo-EM structure[9].

However, the energy difference between the conformations is not large. It is therefore conceivable that alterations in gyrase structures of different bacterial species, or structural differences between the two targets - gyrase and topoisomerase IV – induce a switch of the preferred conformation. Both factors may contribute to the pronounced differences in the antibiotic

activities between cystobactamid analogues, with potency landscapes across the bacterial spectrum that differ in shape rather than just in magnitude.

In summary, the following conclusions can be drawn for the SAR at ring D (Fig. 9). The natural 2-hydroxy-3-alkoxy substitution of the D-ring turned out to be optimal. In contrast, the amide linking rings C and D was amenable to modifications, e.g., with triazols (reported before), amines, or preferably with a reversed amide as in compound **42**. The unique salicylamide structure on ring D is essential for the broad-spectrum activity of cystobactamids, and optimization efforts at ring D should focus on the CD-amide and the ether group.

## AlbD resistance

A prominent principal resistance mechanism for antibiotics is detoxification of the active substance by enzymatic degradation to inactive metabolites. With regard to oligoarylamides, the amidase AlbD has been found to

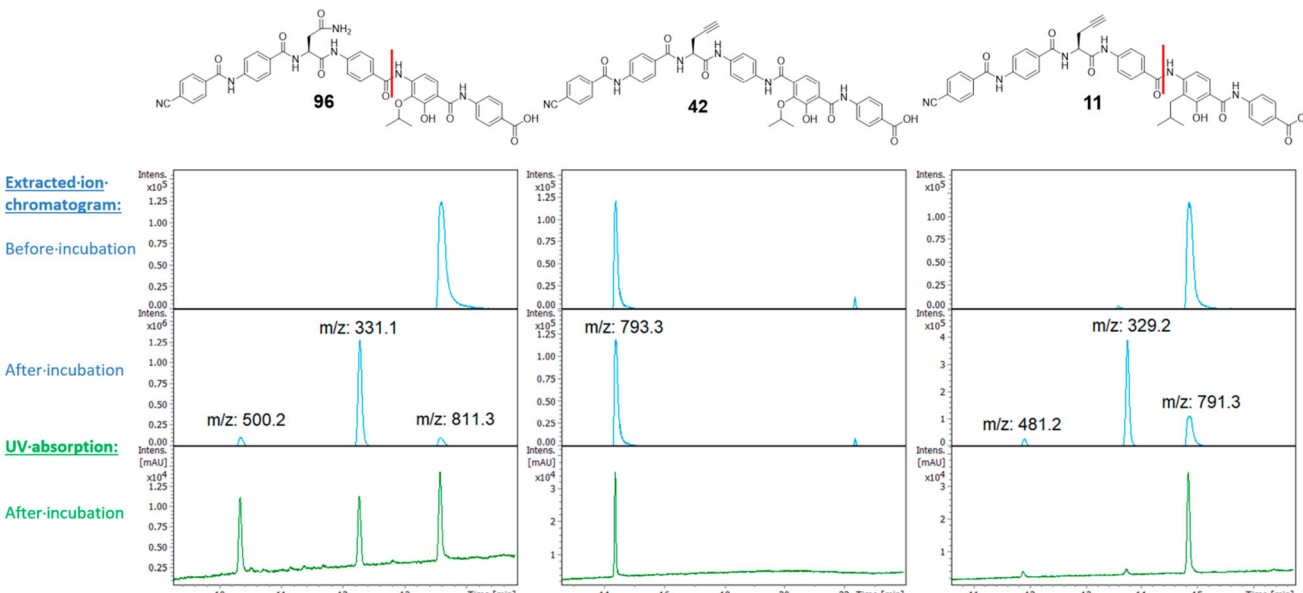

confer bacterial resistance against albicidins[41] by cleaving the CD-amide[42]. AlbD was also found to inactivate cystobactamids[43–45]. The incorporation of a triazol as an isosteric group for the CD-amide could prevent degradation, though[10]. Additionally, metagenome-studies revealed that introduction of a hydroxy function at ring C in *ortho* position to its carbonyl function might decrease susceptibility against endopeptidases such as AlbD[20].

We were interested whether a mere reversion of the CD-amide as in **42** would be sufficient to break AlbD resistance. In addition, we wondered whether the reduction of electron density in ring D, as realized by replacing the oxygen in the isopropoxy group by a methylene group as in **11**, might increase the propensity of the molecule towards proteolytic degradation by AlbD. This was tested with an in vitro assay with recombinant AlbD using our previously established protocol. We included **96** (previously named CN-DM-861) as a validated positive control for the enzymatic reaction[10,42]. This compound is a demethoxy derivative of **1**, has a comparable activity pattern in the MIC assay and was available in sufficient quantities for the assay.

While **96** was degraded after incubation overnight, **42** remained fully stable (Fig. 10). Thus, a reverse amide is no longer recognized as a substrate by the enzyme. Surprisingly, the substitution of the isopropoxy moiety with isobutyl in compound **11** reduced cleavage by AlbD. This suggests that oxygen as HBA might play a role in the active site of AlbD. However, the residual sensitivity of **11** to AlbD indicates that the AlbD insensitivity of **42** is not due to the altered CAA.

## ADME assessment

To get a first understanding of potential ADME-structure properties, we assessed a selection of active as well as inactive derivatives, such as 4, 7, 8, 11 and 42 for their plasma protein binding, plasma stability and metabolic stability. All tested derivatives exhibited a very high plasma protein binding (PPB) in both tested species (mouse and human, Table S4). Although only the free fraction is thought to be bioactive, previously tested analogs such as CN-DM-861 with similar PPB (98.5%) exhibited high in vivo efficacy[10]. Next, we determined the metabolic stability using mouse and human microsomes and found that all tested compounds were fully stable. Finally, the stability in mouse and human plasma was assessed. Again, no liability was observed, because all compounds had half-lifes >240 min (Table S4).

**Fig. 9 | Main SAR findings of the cystobactamid ring D section.** Hydrogen bonds are indicated with dashed red lines, and investigated functional groups are marked with blue circles. All groups were found to be important for activity, and a clear improvement was reached by varying the CD linker.

**Fig. 10 | Comparison of stabilities of synthesized and previous cystobactamids towards AlbD.** Data for previously described **96** (left, retention time: 13.8 min) as positive control[10,42], **42** (middle, ret. time: 14.3 min) and **11** (right, ret. time: 14.9 min) are shown. Reaction products following incubation with AlbD overnight were monitored by LC/MS/UV. The extracted ion chromatograms are depicted for the corresponding masses of cystobactamids and the expected AlbD cleavage products. Cleavage sites are marked red in the structures. The masses 331.1 and 329.2 correspond to the DE-fragments (free amines), the masses 500.2 and 481.2 correspond to the residual A-C fragments (carboxylic acids). For the latter, the UV absorption (190–400 nm) is depicted as well. The full elugrams are found in the SI (Figs. S2–S4).

**Table 3 | Pharmacokinetic properties of 42 in mice**

|  | 42 |
|---|---|
| t1/2 [h] | 0.88 ± 0.1 |
| C0 [μg/mL] | 2.2 ± 1.5 |
| AUC 0-t [ng/mL*h] | 1207 ± 832 |
| Vz [L/kg] | 1.39 ± 1.0 |
| Cl [mL/min/kg] | 17.90 ± 12.3 |

Pharmacokinetic parameters after 1 mg/kg IV dose of **42**. T1/2: half-life, C0: concentration at t = 0 h, AUC 0-t: exposure from 0-t; Vz: volume of distribution, Cl: plasma clearance.

## Pharmacokinetic characterization of 42

As cystobactamid **42** was the most potent derivative and showed favourable ADME in vitro properties, its pharmacokinetic properties were assessed in CD-1 mice after intravenous administration of 1 mg/kg. Compound **42** had a plasma half-life of 0.88 h in mice (Table 3 and Fig. S1).

The initial concentration C0 was around 2.2 μg/ml. The volume of distribution was around 1.4 l/kg, indicating a good distribution into tissue with a low clearance as expected from the in vitro ADME data. Thus, the initial PK data indicate that 42 exhibited favorable properties for further preclinical development.

## Conclusions

This work addressed the synthetically challenging ring D of cystobactamids to investigate structure-activity relationships. Various newly developed synthetic protocols led to 19 analogues, that were evaluated on a panel of bacterial pathogens that including challenging, multidrug resistant strains. We found that the salicylamide structure on ring D is essential for the broad-spectrum activity of cystobactamids and probably constitutes the key pharmacophore of the oligoarylamides. The amide moiety linking rings C and D was amenable to modifications, that led to the reversed amide **42**. The terephthalic acid **42** is highly and broadly active against resistant pathogens and stable against the resistance factor AlbD. First ADME and pharmacokinetic data suggest that **42** has promising properties as an antibiotic lead compound.

## Methods

Details of the experimental procedures and the starting materials used as well as spectra are listed in the supplementary information.

## Data availability

The authors declare that the data supporting the findings of this study are available within the paper and its Supplementary Information files. Should any raw data files be needed in another format they are available from the corresponding author upon reasonable request.

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

## Acknowledgements
The authors thank Dr. Stefan Saretz, Dr. Dominik Heimann, Dr. Matthias Göhl, Mila Le, Norman Birke for synthetic help and advice. We also thank Ulrike Beutling for HRMS measurements, Christel Kakoschke, Esther Surges and Dr. Kirsten Harmrolfs for NMR measurements. We thank Kimberley Vivien Sander, Janine Schreiber and Jennifer Wolf for technical assistance. Funding The studies were co-funded by the German Centre for Infection Research (DZIF; Grant no TTU09.710) and the BMBF Project "Wirkstoffentwicklung auf Basis von Naturstoffen zur Bekämpfung von Infektionskrankheiten" (no GGNATM27).

## Author contributions
M.S., D.K., C.L., and T.S. designed and synthesized cystobactamids. J.C., J.H., and K.C. performed MIC-assays. M.S. and H.F. designed and performed AlbD-related experiments. K.R. performed and analyzed data of ADME and PK studies. A.K., R.M., and M.B. supervised the study. M.S. and M.B. wrote the manuscript. All authors analyzed the data and revised and approved the manuscript.

## Funding

## Competing interests
M.S., M.B., D.K., A.K., T.S., R.M., J.H., K.C., J.C., and K.R. are coinventors on patent applications on natural and synthetic cystobactamids. All other authors declare no competing interests.

## Ethics
Use of research animals or human research participants: For plasma stability experiments, samples were procured from commercial sources. Human plasma was obtained from Antibodies-online (catalog no. ABIN5706569), and mouse plasma was obtained from Tebubio (catalog no. D408-04-0050). The methods were performed in accordance with relevant guidelines and regulations and approved by the Helmholtz Centre for Infection Research. The animal studies were conducted in accordance with the recommendations of the European Community (Directive 2010/63/EU, 1st January 2013). All animal procedures were performed in strict accordance with the German regulations of the Society for Laboratory Animal Science (GV-SOLAS) and the European Health Law of the Federation of Laboratory Animal Science Associations (FELASA). The PK study was approved by the ethical board of the Niedersächsisches Landesamt für Verbraucherschutz und Lebensmittelsicherheit, Oldenburg, Germany.
