## [Transparent Peer Review file · Communications Chemistry]

Synthetic studies on the tetrasubstituted D-ring of cystobactamids lead to potent terephthalic acid antibiotics

Corresponding Author: Professor Mark Broenstrup

Version 0:

Reviewer comments:

Reviewer #1

(Remarks to the Author)

Prof. Bronstrup, Muller and colleagues initially discovered cystobactamid family of antibiotics in 2014 and have since reported many important studies on this compound class. In my view, this excellent manuscript significantly advances our understanding of cystobactamids. The study provides new insights into the structure-activity relationship and identifies a promising lead (compound 42) for further development.

Briefly: the authors synthesized 19 new cystobactamid analogues, focusing on central D-ring and adjacent amide bonds. The natural 2-hydroxy-3-alkoxy substitution pattern on the D-ring was optimal for broad-spectrum activity. The amide between rings C and D was amenable to modification. Reversing this amide bond (compound 42) led to excellent broad-spectrum antibiotic activity. Compound 42 was resistant to degradation by AlbD; a known resistance mechanism against this class. Initial PK studies on compound 42 in mice look promising. Compounds were evaluated against a wide panel of bacterial strains, including challenging multidrug-resistant pathogens. The manuscript includes much complex synthetic chemistry, with procedures well reported and compounds well characterized.

Overall, the authors present a coherent and detailed study on the structure-activity relationships of cystobactamid antibiotics. Their conclusions appear well-supported by the experimental data presented. I congratulate the authors on the quality of this work and recommend publication. I have only very minor suggestions below.

Suggestions:

- compound 96 is mentioned in the AlbD resistance section without being previously introduced or defined (unless I missed it)
- I don't understand the meaning of "AS" in the sentence "...AlbD insensitivity of 42 is not due to the altered central AS." on page 12.
- In the Results section, "conspicuously" should be "conspicuously"

Reviewer #2

(Remarks to the Author)

The continuing and alarming increase in antimicrobial resistance (AMR) has provoked considerable global discourse as to action required to address this issue. A unanimous key recommendation is further research and development to obtain novel antibiotics although pharma has reluctantly declined to invest in such pathways due to the difficulty with countering AMR. Consequently, the onus has largely rested upon academia to undertake such studies. Numerous novel chemical and biological strategies have been developed in the past 24 months, some aided by interesting artificial intelligence analyses, leading to some confidence that new antibiotics may eventuate. In this comprehensive study, the authors have applied an extensive and systematic chemical study on the role of the D-ring within a hexapeptide scaffold of cystobactamids, derived from myxobacteria, that is acknowledged to be critical for antimicrobial activity. The ring, a para-aminobenzoic acid moiety, was subjected to extensive judicious modification and substitution in a structure-activity analysis with the aim of integrating these into the scaffold with a view to identify improved antibacterial agents. Impressive organic chemistry was employed to develop no less than 19 analogues that were then evaluated for activity against a pool of ESKAPE pathogens. Only one analogue, 42, showed potent activity, this being a terephthalic acid with a reversed amide bond between the C and D rings. Subsequent assessment showed this analogue to possess increased pharmacokinetic stability as well as resistance to AMR development. This, together, constitutes an important example of an intricate, laborious, detailed and yet systematic SAR study leading to a "hit" lead molecule for further evaluation as a potential clinically useful antibiotic.

The quality of the chemistry employed within this extensive project is excellent and provides important directions for the development of similar compounds such as those derived from the albicidins, let alone the remaining rings within the cystobactamids. My queries are as follows:

1. What were the overall yields of the each acquired compounds?
2. Were the compounds quantified before antibacterial assay? This is critical given that the variation in polarity between compounds is large and this can significantly influence the level of residual salt retention and, in turn, affect the outcome of MICs.
3. Modelling of 42 interaction with gyrase would be useful (but not essential) in highlighting the role of the modifications in enhanced antibacterial activity.

Overall, this manuscript is well-written, the chemistry excellent and the discussion thorough and well-considered. The SI is extensive and has all the necessary information to provide additional support of the stated conclusions.

Minor editorial corrections:

- p. 1, abstract, line 6: Replace "bioisosters" with "bioisosteres".
- p. 1, column 2, line 4: Replace "was" with "were".
- p. 3, column 1, line 6: Replace "isosters" with "isosteres" (as per heading on p. 4, column 1).

Version 1:

Reviewer comments:

Reviewer #2

(Remarks to the Author)

I am satisfied that the recommended revisions and editings have been fully undertaken. I agree with the comments related to quantitation of the compounds prior to assay. Overall, this is now a fine manuscript that is clearly acceptable for publication in Communications Chemistry.

Point-by-point response

The original comment is printed in black. The author's response is printed in blue, and changes to the text are highlighted with a yellow background.

Reviewer #1 (Remarks to the Author):

Profs. Bronstrup, Muller and colleagues initially discovered cystobactamid family of antibiotics in 2014 and have since reported many important studies on this compound class. In my view, this excellent manuscript significantly advances our understanding of cystobactamids. The study provides new insights into the structure-activity relationship and identifies a promising lead (compound 42) for further development.

Briefly: the authors synthesized 19 new cystobactamid analogues, focusing on central D-ring and adjacent amide bonds. The natural 2-hydroxy-3-alkoxy substitution pattern on the D-ring was optimal for broad-spectrum activity. The amide between rings C and D was amenable to modification. Reversing this amide bond (compound 42) led to excellent broad-spectrum antibiotic activity. Compound 42 was resistant to degradation by AlbD; a known resistance mechanism against this class. Initial PK studies on compound 42 in mice look promising. Compounds were evaluated against a wide panel of bacterial strains, including challenging multidrug-resistant pathogens. The manuscript includes much complex synthetic chemistry, with procedures well reported and compounds well characterized.

Overall, the authors present a coherent and detailed study on the structure-activity relationships of cystobactamid antibiotics. Their conclusions appear well-supported by the experimental data presented. I congratulate the authors on the quality of this work and recommend publication. I have only very minor suggestions below.

We thank you for the positive valuation of our work.

Suggestions:

- compound 96 is mentioned in the AlbD resistance section without being previously introduced or defined (unless I missed it)

We have added the following test to introduce this compound:

We included **96** (previously named CN-DM-861) as a validated positive control for the enzymatic reaction.^{11, 43} This compound is a demethoxy derivative of **1**, has a comparable activity pattern in the MIC assay and was available in sufficient quantities for the assay.

- I don't understand the meaning of "AS" in the sentence "...AlbD insensitivity of 42 is not due to the altered central AS." on page 12.

Thank you for spotting this. CAA is the correct description, AS was the German abbreviation. We corrected it.

- In the Results section, "conspicuously" should be "conspicuously"

Corrected

Reviewer #2 (Remarks to the Author):

The continuing and alarming increase in antimicrobial resistance (AMR) has provoked considerable global discourse as to action required to address this issue. A unanimous key recommendation is further research and development to obtain novel antibiotics although pharma has reluctantly declined to invest in such pathways due to the difficulty with countering AMR. Consequently, the onus has largely rested upon academia to undertake such studies. Numerous novel chemical and biological strategies have been developed in the past 24 months, some aided by interesting artificial intelligence analyses, leading to some confidence that new antibiotics may eventuate. In this comprehensive study, the authors have applied an extensive and systematic chemical study on the role of the D-ring within a hexapeptide scaffold of cystobactamids, derived from myxobacteria, that is acknowledged to be critical for antimicrobial activity. The ring, a para-aminobenzoic acid moiety, was subjected to extensive judicious modification and substitution in a structure-activity analysis with the aim of integrating these into the scaffold with a view to identify improved antibacterial agents. Impressive organic chemistry was employed to develop no less than 19 analogues that were then evaluated for activity against a pool of ESKAPE pathogens. Only one analogue, 42, showed potent activity, this being a terephthalic acid with a reversed amide bond between the C and D rings. Subsequent assessment showed this analogue to possess increased pharmacokinetic stability as well as resistance to AMR development. This, together, constitutes an important example of an intricate, laborious, detailed and yet systematic SAR study leading to a "hit" lead molecule for further evaluation as a potential clinically useful antibiotic.

The quality of the chemistry employed within this extensive project is excellent and provides important directions for the development of similar compounds such as those derived from the albicidins, let alone the remaining rings within the cystobactamids. My queries are as follows:

1. What were the overall yields of the each acquired compounds?

We added the yields and the number of steps to the head of the MIC-tables.

2. Were the compounds quantified before antibacterial assay? This is critical given that the variation in polarity between compounds is large and this can significantly influence the level of residual salt retention and, in turn, affect the outcome of MICs.

The amount of salt was minimized by prolonged lyophilization time after the isolation of the final product. Main characterization by NMR, nevertheless salt residues of ammonium bicarbonate may be present, but should be below 10 wt%. We assume this on the basis of the reaction conversion achieved (monitored by LCMS) and the chromatogram itself. This uncertainty is much smaller compared to the uncertainty of the microbiological assays, which are in the range of 1 dilution step (i.e. half or double concentration).

3. Modelling of 42 interaction with gyrase would be useful (but not essential) in highlighting the role of the modifications in enhanced antibacterial activity.

A modelling was tried on gyrase structures, but we did not obtain conclusive results. We therefore focused more on a rational discussion of activity relevant conformations. In a follow-up paper, we

will provide additional structural insights based on Cryo-EM structures of cystobactamids; however, these studies are yet to be completed.

Overall, this manuscript is well-written, the chemistry excellent and the discussion thorough and well-considered. The SI is extensive and has all the necessary information to provide additional support of the stated conclusions.

We thank you for the positive valuation of our work.

Minor editorial corrections:

p. 1, abstract, line 6: Replace "bioisosters" with "bioisosteres".

Corrected

p. 1, column 2, line 4: Replace "was" with "were".

Corrected

p. 3, column 1, line 6: Replace "isosters" with "isosteres" (as per heading on p. 4, column 1).

Corrected